# STAT3 and Its Pathways’ Dysregulation—Underestimated Role in Urological Tumors

**DOI:** 10.3390/cells11193024

**Published:** 2022-09-27

**Authors:** Maciej Golus, Piotr Bugajski, Joanna Chorbińska, Wojciech Krajewski, Artur Lemiński, Jolanta Saczko, Julita Kulbacka, Tomasz Szydełko, Bartosz Małkiewicz

**Affiliations:** 1Department of Minimally Invasive and Robotic Urology, University Center of Excellence in Urology, Wrocław Medical University, 50-556 Wroclaw, Poland; 2Department of Molecular and Cellular Biology, Faculty of Pharmacy, Wroclaw Medical University, 50-556 Wrocław, Poland; 3Department of Urology and Urological Oncology, Pomeranian Medical University, Powstańców Wielkopolskich 72, 70-111 Szczecin, Poland

**Keywords:** STAT3, prostate cancer, bladder cancer, upper tract urothelial carcinoma, renal cell carcinoma, penile cancer, testicular cancer

## Abstract

Nowadays, molecular research is essential for the better understanding of tumor cells’ pathophysiology. The increasing number of neoplasms is taken under ‘the molecular magnifying glass’; therefore, it is possible to discover the complex relationships between cytophysiology and tumor cells. Signal transducer and activator of transcription 3 (STAT3) belongs to the family of latent cytoplasmic transcription factors called STATs, which comprises seven members: STAT1, STAT2, STAT3, STAT4, STAT5A, STAT5B, and STAT6. Those proteins play important role in cytokine-activated gene expression by transducing signals from the cell membrane to the nucleus. Abnormal prolonged activation results in tumorigenesis, metastasis, cell proliferation, invasion, migration, and angiogenesis. Inhibition of this transcription factor inhibits the previously mentioned effects in cancer cells, whereas normal cells are not affected. Hence, STAT3 might be a viable target for cancer therapy.

## 1. Introduction

All urological cancers made up 13% of incidence and 8% of mortality of all cancers in 2020 worldwide. PCa is one of the most common cancers in men and the most common cancer in urology, representing about 56% of all urological cancers in 2020 [1]. The most important risk factors are age (the majority of PCa diagnosis occurs between the age 65 and 74) and familial history of PCa [2]. Bladder cancer is the second-most-common and deadly cancer found and diagnosed in urology, causing death in men approximately four times more than in women. The known risk factors are smoking, exposure to aromatic amines, chronic urinary tract infections, pelvic radiotherapy, or cyclophosphamide chemotherapy [1,3]. RCC is third in both incidence and mortality among urological cancers, representing about 17% and 23%, respectively [1]. Smoking, hypertension, and obesity predispose to its occurrence [4]. UTUC is a rare tumor, with an incidence estimated at 1–2/100,000. The major risk factors are similar to those of RCC [5]. PeCa is another occasional condition, accounting for 0.2% incidence and 0.1% mortality of all cancers, which is described as epidemiologically significant mainly in the countries of South America and Africa [1,2,3,4,5,6]. Similar to PeCa, testicular cancer is one of the most uncommon cancers. It is frequently diagnosed in young men, with the highest rates between the ages of 25–29 and 30–43 [1,2].

The STAT protein family includes STAT3 [7]. In normal conditions, STAT3 is latently located in cytoplasm. However, there are many pathways that may activate STAT3, and the major one is the IL-6 pathway. After stimulation, STAT3 phosphorylation, dimerization, and nuclear translocation occur. In the nucleus, the protein acts as a transcription factor, resulting in enhanced cell survival, proliferation, migration, and angiogenesis and inhibited apoptosis [8].

STAT3 has already been described as a prooncogenic protein in many different tumors such as those from breast, head and neck, lung, gastric, and pancreatic cancer [9,10]. Moreover, the relationship between not only STAT3 and cancerous cells but also STAT3 and TME has been shown many times, as well as in the case of urological cancers. This process may include changes in the MDSCs’ or TAMs’ phenotype or the differentiation of MSCs into osteoblasts in PCa bone metastasis [11,12,13]. On the other hand, some studies reported suppressing properties of STAT3 activation, which prevent disease recurrence or metastasis. Therefore, we encourage careful use of STAT3 and IL-6 inhibitors [14].

Considering the epidemiological importance of urological cancers as well as multidimensional STAT3 activity in tumor progression and metastasis, we strongly believe that STAT3 is underestimated in urologic oncology. Therefore, we consider this issue worth describing. The aim of this work is to gather and discuss the major recently published findings in the field of urological cancers, through the prism of STAT3 used as a biomarker or therapeutical target, to help researchers when revising their views and exploring this matter.

## 2. Role of STAT3 in Cancers

STAT3 belongs to the family of latent cytoplasmic transcription factors called STATs, which comprises seven members: STAT1, STAT2, STAT3, STAT4, STAT5A, STAT5B, and STAT6. Those proteins play important role in cytokine-activated gene expression, by transducing signals from the cell membrane to the nucleus [15,16].

STAT3 is characterized by six main structural motifs: the amino-terminal domain, coiled-coil domain, DNA binding domain, linker domain, SH2 domain, and transactivation domain [16]. The SH2 domain is the most specific part of the protein. It is responsible for the identification and binding of phosphotyrosine motifs. Furthermore, it provides recognition and binding by JAK protein activation. Finally, it allows dimerization of STAT3, either with another STAT3 molecule or with the remaining STAT family members [17]. In regular conditions, STAT3 is in a latent state in the cytoplasm, after the external signals of the cell surface receptors oligomerize. which leads to the proximation of the tyrosine kinases and triggers their transphosphorylation, finally resulting in the activation of the JAK kinases. Further phosphorylation of the internal domain of the receptor, by the membrane receptor kinases and non-receptor cytoplasmic kinases, including ABL or Src-related kinases, ends up with the recruitment of STAT3 and its phosphorylation. Activation of STAT3 causes either the homodimerization or heterodimerization of protein and translocation to the nucleus, where it stimulates gene expression by binding to DNA [18]. The JAK/STAT3 pathway is immediately quenched by SOCS, PIAS, and PTPases or through protein degradation by ubiquitin–proteosome machinery [8]. STAT3 is activated mainly by IL-6 and EGF; however, many other factors have been exposed as taking part in this process [19]. STAT3 pathway activation is presented in Figure 1.

Upon binding to DNA, STAT3 regulates the expression of many important genes such as the Bcl-2, Bcl-xl, Bcl-6, and survivin responsible for cell survival and the inhibition of apoptosis; the MYC and cyclin D1 regulators of proliferation; the MMPs’ promotors of metastasis and migration; the VEGF mediator of angiogenesis; and the IL-6 pro-inflammatory cytokine and IL-10 immunosuppressive cytokine. Interestingly, the transcriptional effect of STAT3 varies among tissue types [18]. Aberrant phosphorylation of STAT3 was reported in 70% of cancers and was associated with poor prognosis [18]. The aforementioned refers not only to solid tumors but also to hematological malignancies.

Constitutive activation of this transcription factor was detected previously in acute myeloid leukemia, multiple myeloma, non-Hodgkin’s lymphoma, and chronic lymphocytic leukemia as well as solid tumors of the lung, breast, ovary, cervix, prostate, bladder, kidney, colon, liver, stomach, and head and neck [18]. The aberrant activation of STAT3 might be caused by spontaneous mutation of protein; however, it is mostly assigned to autocrine and paracrine cytokine stimulation [7,18]. Additionally, hyperactivation of STAT3 was observed as a result of MEK therapeutical inhibition [20]. Abnormal prolonged activation results in tumorigenesis, metastasis, cell proliferation, invasion, migration, and angiogenesis. Inhibition of this transcription factor inhibits the previously mentioned effects in cancer cells, whereas normal cells are not affected. Hence, STAT3 might be a viable target for cancer therapy [21].

## 3. Role of STAT3 in Prostate Cancer

### 3.1. Description and Epidemiology of Prostate Cancer

In 2020, prostate cancer was the second-most-frequent cancer in men worldwide. It was also the fifth-highest cause of cancer death in this group. Incidence has the highest rate in Northern and Western Europe, the Caribbean, and Australia/New Zealand [1]. However, a sharp reduction in prostate cancer incidence between 2010 and 2014 was reported. The five-year relative survival rate of localized and regional PCa is estimated to be >99%, whereas in distant cancer the rate dramatically decreases to 30% [22]. This may indicate why scientists put so much effort into developing new strategies for managing PCa. It was also shown that Africans are more susceptible to develop PCa during their life, while Asians are less affected. Such differences have yet to be explained [23].

### 3.2. Why Is STAT3 Crucial in Prostate Cancer Development and Progression?

Plenty of research papers have highlighted the importance of STAT3 in PCa development. Many factors and different conditions may trigger STAT3 activation, but the core seems to be the NF-κB/IL-6/STAT3 axis [24,25,26,27,28]. There are also lots of downstream targets under the control of STAT3. Overexpressed STAT3 forces cyclin D1, Bcl-2, Bcl-xL, and survivin expression, which have proliferative and antiapoptotic properties [29,30,31]. PCa cells may also avoid apoptosis by inhibited expression of caspase 3 and caspase 9 through STAT3 activity [32]. On the other hand, elevated levels of N-cadherin, vimentin, MMP2, and MMP9 (mesenchymal factors) and a decreased level of E-cadherin (epithelial marker) enhance the invasive and migrative capabilities, leading to EMT [26,33,34]. Overexpression of VEGF results in augmented angiogenesis [35]. Lastly, induced expression of PD-L1 helps a tumor to avoid a physiological immune response [36]. Collectively, high expression and activation of STAT3 result in increased cell survival, proliferation, angiogenesis, invasion, and migration, leading to distal metastasis [37,38,39]. STAT3 signaling pathways, their regulators, their downstream targets, and the effects of their activation are shown in Figure 2.

### 3.3. TME Is Crucial for Prostate Cancer Progression

Tumors might be described as complicated organs that consist of not only tumorous cells but also benign cells, e.g., stromal cells. All of these create TME, which has recently caught scientists’ interest. Exemplary stromal cells are fibroblasts—critical regulators of metastatic progression in PCa. PLZF takes part in self-renewal or stem cell differentiation and may act as a tumor suppressor. Collected data showed that PLZF level is decreased, whereas pSTAT3 level is increased, with PCa progression. Exogenous overexpression of PLZF resulted in substantial inhibition in STAT3 phosphorylation by increased SHP1 expression, which has the ability to deactivate JAK/STAT3 pathway. On the other hand, fibroblasts produce CCL3, which prevents PLZF expression. Scientists presented a part of the complicated relationship between fibroblasts and PCa cells, describing a potentially useful CCL3/PLZF/SHP1/STAT3 cascade [40].

Zhao et al. reported that highly concentrated LTF significantly decreases STAT3 and GM-CSF levels, which results in immune TME changes [41].

One of the most abundant immune cell populations in TME are TAMs. CCL5 is a chemokine produced by TAMs. An elevated CCL5 level was noticed in PCa tissues and has been associated with migration, invasion, and EMT promotion. CCL5 promotes self-renewal of PCSCs by activating their CCR5 and, thereby, stimulating the β-catenin/STAT3 signaling pathway, which results in STAT3 upregulation. In vivo studies revealed that CCL5 blockage leads to the inhibition of PCa growth, bone metastasis, and PCSCs self-renewal. Therapy focused on inhibiting TAMs or CCL5 and CCR5 might be a promising future approach [33]. TAMs may be divided into two groups—M1-like, which acts as anticancer cells, and M2-like, which promotes cancerous characteristics. One phenotype can switch to the other, under specific conditions. It was evidenced that the use of PC3 supernatant results in the disappearance of M1-like biomarkers and the emergence of M2-like biomarkers. Additional STAT3 inhibitor completely changes this trend and forces differentiation to the M1-like phenotype. Modifying TME by TAMs’ manipulations may contribute to develop new strategies [42]. Another study focused on exploring the role of IL-8 secreted by M2-like biomarkers. This showed that IL-8 activates the STAT3/MALAT1 tissues pathway and, therefore, promotes PCa progression. MALAT1 gene knockdown results in the inhibition of proliferation and invasion. Whether STAT3 activation is pivotal for MALAT1 overexpression still has to be explored [43].

Another part of TME are MDSCs. MDSCs characterized as CD33^+^ pSTAT^+^ are more frequent in PCa TME, compared to BPH tissues. This described phenomenon may be the basis for future studies on developing new treatment strategies [11]. MDSCs play a crucial role in suppressing antitumor immunity, and their generation may be induced by PCa cells. Galiellalactone, a STAT3 inhibitor, effectively downregulates MDSCs’ frequency. Researchers conclude that galiellalactone may impair arising antitumor immunity, suggesting a potential use for STAT3 inhibitors in advanced PCa [44]. An innovative approach to change TME in the favor of patients may come with the combined treatment of STAT3 inhibition and TLR9 stimulation. The developed therapy significantly changed TME, via neutrophiles and CD8+ T cells recruitment, and decreased the MDSCs’ population at the same time. The results suggest that a bifunctional combination has the potential to disrupt the tumor immunological escape mediated by STAT3 [45].

McGuire et al. described a process of MSCs recruitment by PCa bone metastasis. Firstly, MSC-derived IL-28 leads to PCa cells’ apoptosis, although persistent exposure results in the selection of cells resistant not only to IL-28-induced apoptosis but also to chemotherapeutics such as docetaxel or etoposide. Scientists reported that the use of a STAT3 inhibitor, S32-201, selectively inhibited MSC-selected PCa cells. Such cells are notably susceptible to STAT3 inhibition in vivo [13].

Witt et al. analyzed how the use of STAT3 inhibitor will influence anti-CTLA-4 treatment. Cotreatment resulted in a significant enhancement in the survival time of mice, when compared to the group treated with antibody only. However, complete tumor regression was not observed. Significant CD45^+^ (characteristic for all leucocytes) cells’ tumor infiltration and a substantial reduction in regulatory T cells’ population (which may contribute to the tumor’s resistance development) were noticed intratumorally. We point to the therapeutical potential of such an approach [46].

Apoptotic cells are supposed to produce paracrine molecules that promote a compensatory proliferation mechanism among survived cells. Conducted experiment showed that after etoposide-induced apoptosis, PC3 repopulation, EMT, and chemoresistance occur through the caspase-3/cPLA2/COX-2/PGE-2/EP4/2/STAT3 axis, called Phoenix Rising. Although Phoenix Rising is a physiological process responsible for regeneration in healthy tissues, some epigenetic-induced changes may lead to pathological repopulation of cancerous cells. New insight into the repopulation mechanism may result in new therapeutical strategies [47].

### 3.4. Attempts at Breaking CRPC

CRPC often remains lethal or refractory for available therapies. Many researchers consider developing tools to break this resistance as key for saving PCa patients. Attempts have been made to use NK cells to manage CRPC. IL-6-producing tumors were shown to be more resistant to NK cell cytotoxicity, suggesting the importance of IL-6 signaling in determining tumor cell sensitivity. Moreover, high expression of IL-6 led to over-expression of PD-L1 in CRPC cells, which, in turn, resulted in the death or inactivation of T cells and decreased expression of the NKG2D ligand, which disrupted NK cells in recognizing cancer cells. Combination therapy, with a JAK or STAT3 inhibitor with the PD-L1 antibody, resulted in a significant decrease in PD-L1 levels and an increase in NKG2D levels as well as the increased susceptibility of CRPC cells to cytotoxic NK cells. [36]. It was also presented that FBP1, a glycolysis inhibiting enzyme, acts through the STAT3 pathway, and its loss leads to increased expression of PD-L1. Research revealed that FBP1, independently of its enzyme activity, interferes with STAT3 and inhibits STAT3 binding to the locus of the PD-L1 gene. This effect was reversed by ionizing radiation or IL-6 administration, which increased STAT3 phosphorylation. Knockdown of the FBP1 gene led to a significant enhancement of the PD-L1 protein, mRNA expression, and tumor growth. Interestingly, loss of the FBP1 gene correlated with stronger resistance to anti-PD-L1 treatment. Collectively, FBP1 loss may be involved in the tumor immune escape. Further studies are needed to define new strategies [48].

HepaCAM is an immunoglobulin-like molecule poorly expressed or absent in malignant tumors. Recent study showed HepaCAM interferes with the IL-22/STAT3 axis, consequently blocks STAT3 phosphorylation, and noticeably decreases proliferation and the migration and invasion of CRPC cells. Results demonstrated not only reduced levels of the STAT3 target genes but also the absence of any lung metastatic areas in mice. Restoring expression of HepaCAM might be a promising perspective for CRPC patients [30].

Lin et al. reported that CYP1B1, an enzyme catalyzing the synthesis of 4-OHE2 from estradiol, might contribute to the development of CRPC by promoting PCSCs’ characteristics. In addition, 4-OHE2 is supposed to increase IL-6 expression, which, in turn, intensifies the IL-6/STAT3 pathway and its downstream genes. Scientists found out that CYP1B1 expression positively correlates with GS and is higher expressed in CRPC tissues, compared to the androgen-dependent PCa cells. Moreover, CYP1B1 enhanced CRPC resistance to bicalutamide, while its knockdown reversed this effect. CYP1B1 seems to be a new therapeutic target in CRPC patients [49].

### 3.5. Breaking the Chemoresistance

Various therapeutic strategies can induce resistance in PCa cells, which may contribute to cancer cell survival. ALT has been shown to reduce the viability of CSC cells, by inhibiting STAT3 and sensitizing these cells to cisplatin [50]. Hu et al. focused on the phenomenon of docetaxel-induced autophagy in CRPC cells. The collected data showed that activated STAT3 reduces the viability of CRPC cells during chemotherapy, by promoting apoptosis. Finding a molecular explanation for this result could be the cornerstone of acquired resistance management [51].

Galiellalactone, a direct STAT3 inhibitor, was reported to significantly decrease docetaxel-resistant PCa cells’ viability and may be used in combined therapy [52]. Similarly, another team proved galiellalactone’s efficacy in managing enzalutamide-resistant PCa, both in monotherapy and in combination [53].

Furthermore, metformin is capable of reversing the EMT promoted by ENZ, by targeting the TGF-β1/STAT3 axis. Combined therapy of metformin and ENZ was especially effective and promising [25]. Although ENZ prolongs PCa patients’ lives by about 5 months, it can also induce NED by activating the lncRNA-p21/EZH2/STAT3 pathway. Developed NEPCs are insensitive to ADT and, therefore, exacerbate the course of illness. Scientists suggest that EZH2 targeting may reduce enzalutamide-induced changes [54].

The collected data implied that STAT3 activation may result in radioresistance. Zhang et al. proved that the use of a STAT3 inhibitor or STAT3 knockdown increases the sensitivity of PCa cells to irradiation. Complex treatment of radio- and chemotherapy (STAT3 inhibition or knockdown) demonstrated a synergistic effect, as shown by augmented apoptosis. These results seem to be an interesting approach to PCa managing, combining two methods of cancer treatment [55].

### 3.6. Targeting NF-κB/IL-6/STAT3 Axis

Recently, some studies focused on targeting NF-κB/IL-6/STAT3 axis have been published. It has been shown that the IL-8 level is elevated in PCa cells, which considerably promotes proliferation as well as migration and invasion, while inhibiting apoptosis. Mechanistically, IL-8 works by activating the STAT3/AKT/NF-κB axis. This finding may contribute to the development of new treatment strategies [56]. Considering the pivotal role of the aforementioned axis in PCa progression, scientists have developed a dual STAT3/NF-κB inhibitor. Iridium(III), showing anti-NF-κB properties, was conjugated with benzofuran, which acts as a STAT3 inhibitor. The collected data shows that a synthesized complex not only inhibits STAT3 activation and the binding of already activated STAT3 to DNA but also decreases the nuclear translocation of NF-κB from the cytoplasm. Interestingly, benzofuran-iridium(III) was relatively more toxic against DU145 (prostate cancer cell line) cells than cisplatin and doxorubicin, with simultaneous lower toxicity to normal human cell lines. This conjugation seems to be an interesting and promising method of treatment [28]. NDRG1 is an important molecular regulator, inhibiting PCa progression and metastasis. It inhibits many precancerous signaling pathways, which promote CRPC development. It has been proven that NDRG1 significantly decreases levels of activated STAT3, IL-6, and NF-κB. Considering that NDRG1 affects crucial steps in the NF-κB/IL-6/STAT3 axis and, therefore, disrupts the androgen-independent AR activation pathways, it might be a promising solution for CRPC patients [57].

### 3.7. Can We Treat Prostate Cancer Patients with RNA?

Some scientific teams have, recently, tried to use RNA molecules as a treatment targeted at STAT3. Wei et al. found out that an lncRNA called MAGI2-AS3 is one of the most downregulated lncRNAs in PCa tissues. MAGI2-AS3 is supposed to act as a sponge for another lncRNA appearing in PCa cells—miR-424-5p—which activates the STAT3 pathway. It has been shown that MAGI2-AS3 forced overexpression decreases STAT3 concentration and, in turn, inhibits cell viability and enhances apoptosis [58]. Another study presented that miRNA-583 expression in PCa tissues is significantly inhibited. miRNA-583 transfection resulted in the considerably diminished proliferation and invasion of PCa cells due to JAK1 inhibition and, as a result, abolished activation of STAT3. The collected data suggest a potential use of miRNA-583 as a treatment. The described mechanism of action has remained unknown until this study [59]. A relationship between specific lncRNAs and TAMs has also been described. It has been proved that LINC00467 induces macrophage polarization from the M1-like phenotype to the M2-like phenotype, which is responsible for STAT3 pathway activation through the miR-494-3p/STAT3 cascade. Potentially, blocking this molecule might benefit patients by inhibiting the proliferation and infiltration of PCa cells [60]. Similar properties to those mentioned before are represented by another lncRNA—LINC00473. This activates the JAK/STAT3 pathway and, therefore, contributes to the proliferation of PCa cells. Consequently, inhibitory targeting of LINC00473 might be a new way of treatment [61].

### 3.8. Importance of the Environment on the Molecular Level

It is commonly known there are many environmental risk factors for developing cancers. The way we live and create our habits might also contribute to PCa emerging. Kwan et al. proved that a high-fat diet increases tumor size, STAT3 phosphorylation, and the PA levels in xenograft tissues. PA upregulates STAT3 mRNA and protein. Moreover, PA strongly binds to STAT3, which changes its conformation and activity [27]. Another team noted that LDL cholesterol significantly increases pSTAT3 level, by enhanced JAK1 and JAK2 phosphorylation. LDL intensifies the proliferative and invasive abilities of PCa cells. The use of statins may benefit Pca patients [29]. The impact of obesity on Pca development was investigated by exploring the role of leptin. It was shown that increased leptin concentration boosts EMT by inducing STAT3 phosphorylation. What is more, it was noticed that the level of leptin receptor is much higher in adenocarcinoma than in BPH [26]. Moreover, 27HC was found to impair lipid rafts and inhibit their signaling pathways. An in vivo experiment showed that 27HC treatment resulted in a statistically significant difference in tumor size in the treated group, when compared to the control group. Mechanistically, 27HC disrupts the IL-6/JAK/STAT3 pathway and, what is more, acts synergistically with STAT3 inhibitors. Further studies are needed to apply these findings clinically [62].

Despite calcitriol’s importance in regulating calcium and phosphorus metabolism, it also has anti-inflammatory and anti-tumor properties. It was revealed that calcitriol inhibits the LPS-induced migration and invasion of PCa cells. Calcitriol enhances the physical interaction between STAT3 and VDR, resulting in the disrupted nuclear translocation of STAT3. Moreover, calcitriol leads to VDR and NF-κB binding, which, in turn, downregulates IL-6 and IL-8. Clinical relevance and in vitro studies have to be carried out [63]. The same pro-cancerous mechanism was analyzed through the prism of melatonin activity. This molecule not only inhibited LPS-induced invasion and migration but also affected not stimulated cells in the same way. The expression of NF-κB as well as the IL-6 and STAT3 levels were noticed to be downregulated. In vivo studies might confirm the therapeutical potential [64].

### 3.9. Different Approaches to Manage PCa

KLF5 is a zinc-finger transcription factor regulating proliferation, apoptosis, and invasion. It was observed that PCa tissues are characterized by the downregulation of KLF5. KLF5 expression is significantly inhibited in tissues described as GS8–GS10, and its concentration is lower in PCa metastasis rather than in localized PCa. That is why scientists assumed that loss of KLF5 might promote invasive abilities of PCa, which, consequently, was proven in experiments. Mechanistically, KLF5 downregulation activated the IGF1/STAT3 pathway, which, in turn, led to PCa invasion. KLF5 was also reported to cooperate with HDAC1 in binding to the promoter of the IGF1 gene [24].

The role of insulin-like growth factor IGF and its receptor was also explored in another study. MDA-9/syntenin is a protein overexpressed in many types of human cancers, and its upregulation caused enhanced invasive abilities in PCa cells. Overexpression of MDA-9/syntenin was observed to correlate with STAT3 overactivation. MDA-9/syntenin physically interacts with IGF1R and leads to its autophosphorylation, which, consequently, activates STAT3. All in all, MDA-9/syntenin-IGF1R interaction is yet to be explored [65].

EA is a diuretic with the potential of inhibiting STAT3 by activating phosphatases SHP2 and PTP1B, which, consequently, dephosphorylate STAT3 at Tyr705. In vivo experiments revealed EA’s antiproliferative properties [66]. Similar effects were previously observed using Capz, a synthetic analogue of capsaicin. Capz reduced STAT3 phosphorylation and nuclear translocation by blocking the phosphorylation of Tyr705. It was discovered that Capz increases the expression of PTPε, which consequently deactivates STAT3. Capz dramatically reduces downstream STAT3 target genes’ levels. The potential involvement of the NF-κB axis in this mechanism is still unknown [67].

The PTMs of STAT3 might not only be useful as a biomarker but also as a potential therapy. Inhibiting STAT3 phosphorylation, acetylation, or glutathionylation at specific amino acids could be a way to disturb the driving of the intracellular signals that are conducting the PCa progression [68].

Commonly used antiandrogens might inhibit PCa’s proliferation, while excessing PCa’s invasive abilities at the same time. It was shown that ASC-J9 is able to induce STAT3 sumoylation, which leads to decreased STAT3 phosphorylation. Adding this result to its previously known ability of degrading AR, ASC-J9 seems to be a promising candidate for supporting ADT or radiotherapy [69].

Virotherapy seems to be an interesting and promising way of managing PCa. In vitro studies showed that NDV induces the ICD markers of PC cells. The addition of STAT3 inhibitor resulted not only in decreased STAT3 phosphorylation but also in significant enhancement of the released ICD markers. The mechanism of this synergistic effect is waiting to be explored [70].

Metformin is a drug widely used for type 2 diabetes. Tang et al. tried to use its potential to inhibit the EMT of the PCa cells induced by ADT. Metformin reduced the migration and invasion by about 50%, which was statistically significant and consistent with the biomarkers’ levels specific for EMT. Finally, it was shown that metformin significantly decreased the pSTAT3 level, without influencing tSTAT3, by inhibiting the COX-2/PGE-2/STAT3 axis. Interestingly, highly concentrated metformin is capable of inhibiting STAT3 directly, even with the presence of exogenous PGE-22 [34].

SAM, which acts as a biological methyl donor, seems to be another promising STAT3 inhibitor. Studies showed a significant reduction of STAT3 protein and its phosphorylated form, after 72 h and 120 h of SAM treatment, respectively [39].

It was demonstrated that cell lines without AR expression show the highest levels of fibrinogen. Knockdown fibrinogen gene, which remains under IL-6/STAT3 control, resulted in the inhibited proliferation and mobility of PCa cells [71].

### 3.10. Should We Pin Our Hopes on Nature?

Lots of molecules have the potential to become an anticancer drug. Fucoidan, a polysaccharide sourced from brown algae, was found to reduce activated JAK and STAT3 levels in PCa tissues, presenting a massive antiangiogenic ability [35]. Atractylenolide II, a natural sesquiterpene lactone, inhibits JAK2/STAT3 pathway activity [72]. Proscillaridin A, a cardiac glycoside obtained from *Urginea maritima*, disrupts the same pathway and, even more importantly, substantially enhances the antiapoptotic abilities of doxorubicin [73]. Liu et al. reported acetyl-11-keto-β-boswellic acid, a pentacyclic triterpenic acid collected from gum resin trees, as a strong cytotoxic agent in PC3 cells resistant to docetaxel. This compound downregulates not only pJAK2 and pSTAT3 but also IGF1R and pAKT levels [74]. On the other hand, carvacrol, a natural flavoring approved for food use, significantly reduces IL-6 and STAT3 expressions, resulting in the limited invasion, migration, proliferation, and viability of PCa cells. The whole mechanistic picture is yet to be explored [75]. There are also some organic molecules and their derivatives that inhibit STAT3 activity through direct binding, preventing STAT3 dimerization and nuclear translocation [76,77,78]. CK, a saponin obtained from ginseng, was reported to increase miR193a-5p expression in the DU145 cell line, which leads to attenuated STAT3 and PD-L1 expression. Thus, CK demonstrates its proapoptotic properties through the lack of T cells’ inactivity [31]. Similarly, PD-L1 inhibition through STAT3 pathway deactivation seems to be possible with the use of CFF-1, a traditional Chinese medicine cure. CFF-1 is likely to act alone or in combination with docetaxel to present its effects. It is reported to inhibit tumor growth and lung metastasis [79]. Methyllucidone, a cyclopentenedione isolated from some Lauraceae family plants’ fruit, has the ability to inhibit STAT3 activation by even 90%. Mechanistically, methyllucidone exerts MEG2 expression, a PTP known for its capability of STAT3 dephosphorylation [80]. On the other hand, Qi Ling, another medication from traditional Chinese medicine, has the potential to alter TME and force TAMs to switch from the M2-like phenotype to the M1-like phenotype through IL-6/STAT3 pathway inhibition. Moreover, Qi Ling is supposed to decrease the paclitaxel resistance in PCa tissues [12]. Furthermore, a polymethoxyflavone obtained from citrus, called nobiletin, was reported to decrease STAT3 expression with consecutive enhancement in bicalutamide cytotoxicity [81]. Astaxanthin, which is naturally produced by marine organisms such as algae, was proven to inhibit proliferation and colony forming by PCa cells, by disrupting STAT3 and related pathways, e.g., JAK2 or NF-κB [32]. The final described product of Chinese medicine, called compound 154, is collected from the skin of giant toads and was evidenced to work as a STAT3 and AR inhibitor. Besides its increased cytotoxicity to normal tissue, it might be a therapeutical option after molecular modifications [82].

### 3.11. Role of STAT3 in Prostate Cancer Diagnostics

Diagnostics is the first step of a long journey of treatment. Well-developed biomarkers characterized by high sensitivity and specificity might act as a powerful weapon in clinicians’ hands. Early and accurate diagnosis yields appropriate treatment, which may provide better outcomes. It seems to be crucial to diagnose benign conditions such as BPH, before their progression to PCa. Sanaei et al. tried to use pSTAT3 as a biomarker of MDSCs—immature cells that accumulate in pathological condition of inflammation and exist in TME. These cells were described as CD33^+^ pSTAT3^+^, and pSTAT3 marking was used to differentiate MDSCs from other myeloid cells, which might act as anticancer factors. The research showed that CD33^+^ pSTAT3^+^ cells were significantly frequent in the patients with PCa, in comparison to the control group with BPH. However, there were no relevance between MDSCs’ level and GS. Researchers concluded that elevated MDSCs level might indicate progression from BPH to PCa [11]. The STAT3 molecule undergoes some specific PTMs. Researchers found out that these alterations are characteristic for different cellular conditions. STAT3 acetylation at Lys685 was observed in overall inflammation, whereas glutathionylation or phosphorylation at Ser727 was more specific for conditions of oxidative stress. What is more, the mentioned PTMs were correlated with GS. Lys685 acetylation was detected in tissues described as GS6. On the other hand, Ser727 glutathionylation or phosphorylation was noticed in GS9. In turn, phosphorylation at Tyr705 was common for all STAT3 signaling pathways. Results suggest that detecting specific STAT3 PTMs might be a biomarker for PCa prevention or differentiation [68]. Marginean at al. assessed the nuclear expression of STAT3 phosphorylated at Tyr705 and Ser727 in the prostate stromal compartment of the cancer and non-cancer areas in hormone-naive patients after a radical prostatectomy due to localized PCa. Lower nuclear expression of STAT3^Tyr705^ and STAT3^Ser727^ in the stromal compartment was observed in cancer tissues, compared with non-cancer tissues. This decreased expression was correlated with shorter time to BCR. Although non-cancer tissues were collected from distant areas of the tumor from patients with PCa, the data have the potential to be the foundation for developing useful biomarkers. The presented evidence reveals similar prognostic power to GS and staging or surgical margin status, which are widely used in early PCa [83]. Similar studies were carried out earlier by another Swedish team, which focused on investigating the expression of tSTAT3 and its phosphorylated forms—STAT3^Tyr705^ and STAT3^Ser727^—in prostate epithelial cells and their impact on disease outcome. Surprisingly, all forms of STAT3 were lower expressed in the cancer cores than in the benign cores, and the lowest expression was detected in the tissues with higher GS. The collected data suggests a correlation between nuclear and cytoplasmic STAT3^Ser727^ and nuclear STAT3^Tyr705^ expression in cancerous tissues and shorter time to BCR. The lower the expression of STAT3, the poorer prognosis of the disease is. However, using gathered outcomes resulted in impairing prognostic values of GS and pT staging. Scientists conclude it might not be a good way to diagnose early stages of PCa [84].

## 4. Role of STAT3 in Bladder Cancer

### 4.1. Description and Epidemiology of Bladder Cancer

Bladder cancer is the most common malignancies of the urinary tract [85]. It is also the 10th-most prevalent cancer in the world, with roughly 573,000 new cases and 213,000 deaths [1]. The incidence varies between geographical regions, and the highest rates are observed in Europe and North America as well as among the male population in Egypt, Syria, Israel, and Turkey. The lowest incidence rates are reported in Sub-Saharan Africa, Latin America, and some Middle Eastern and Central Asian countries [86]. The disorder can occur as NMIBC, MIBC, and a metastatic form of the disease [87]. NMIBC comprises 80% of diagnosed bladder cancer cases and is often associated with FGFR3 mutation [87]. It is estimated that 15% to 20% of NMIBCs progress to MIBCs, in which the neoplasm has advanced beyond the epithelial cell lining and into the muscles [87,88]. 

### 4.2. STAT3 Levels in Bladder Cancer

Recent investigations exposed that STAT3 plays a significant role in the progression of the disease [89]. STAT3 was not only found to be upregulated in 10 types of bladder cancer cell lines but also in invasive bladder cancer tissue samples [90]. Higher values of pSTAT3 were associated with basal bladder cancer, whereas lower values were detected in luminal bladder cancer. Furthermore, the dependence on STAT3 was differentiated following the cancer cell line. The 5637 cell line had the most significant response to STAT3 inhibition. The values of pSTAT3 concurrently increased with the progression of the disease. In addition, MIBC has shown enhanced expression of nuclear STAT3, in comparison with NMIBC [91]. The differences in STAT3 expression were also observed within urothelial cancers. Specimens with papillary patterns demonstrated significantly lower parameters of STAT3 expression than the non-papillary variants [92]. The STAT3 signaling pathways, their regulators, their downstream targets, and the effects of their activation are shown in Figure 3.

### 4.3. Bladder Cancer Cells’ Viability and Apoptosis

Numerous research teams have inspected the role of STAT3 in bladder cancer. The inhibition of STAT3 resulted in a decrease in cell amount, which was assigned to the process of apoptosis. The anti-apoptotic genes Bcl-xL and Bcl-2 and the survivin levels were downregulated in the treated cell lines. The effect was observable in the WH and UMUC-3 cell lines, whereas no change was found in bladder smooth muscle cells. Interestingly, inhibition of the STAT3 not only led to the expression reduction of apoptotic genes but also induced the cleavage of caspases 3, 8, and 9 [89]. Another study also reported that the apoptosis phenomenon was accompanied by expression attenuation of Bcl-xL and Bcl-2 in the T24 cell line [93]. Treatment with Stattic, a STAT3 inhibitor, significantly reduced the weight of the tumor xenografts. The reduction of tumor growth was up to 50%. During Stattic treatment, a reduction of Ki-67 positive cells was also observed [90].

### 4.4. EMT in Bladder Cancer

STAT3 is also implicated in process of EMT in bladder cancer. Blockage of the STAT3 pathway decreased motility and invasiveness, by the inhibition of MMP2 and MMP9 expression [93]. Treatment of bladder cancer cell lines with Tanshinone IIa, an extract derived from *Salvia miltiorrhiza*, contributed to an increase in the epithelial marker E-cadherin level and reduction of mesenchymal markers such as N-cadherin and vimentin. Additionally, transcription regulators of EMT such as SLUG and SNAIL were also downregulated after therapy. This data implied that Tanshinone IIa can suppress the process of EMT. It was established that extract from Tanshinone IIa exerts its action through the inhibition of STAT3 phosphorylation, which ends in the downregulation of CCL2 [94]. The correlation between STAT3 and EMT was spotted during examinations of the IDO1 enzyme. IDO1 was found to be overexpressed in bladder cancer cell lines and tissues. The enzyme is responsible for the breakdown of tryptophan to kynurenine. It was demonstrated that IDO1 may promote EMT through the IL-6/STAT3/PD-L1 pathway. Suppressed expression of IDO1 resulted in decreased proliferation, migration, and invasiveness of bladder cancer cells [95]. Cancer-associated fibroblasts were reported to be involved in the process of bladder cancer progression, through the stimulation of STAT3 phosphorylation. Cancer-associated fibroblasts secreted IL-6, which activated its receptor on bladder cancer cells, subsequently leading to the increased phosphorylation of STAT3 and providing IL-6-activated EMT programming [96]. Certain RNA molecules affect EMT by influencing STAT3 level as well. For instance, miR-4500 was determined to be downregulated in bladder cancer cells. Ectopic expression of miR-4500 stimulated apoptosis, impaired proliferation, and retarded EMT. The study demonstrated that RNA suppressed STAT3 through base pairing with the 3’-untranslated region of STAT3. Interestingly, the phosphorylated STAT3 levels correlated with CCR7, increased expression of miR-4500, and repressed STAT3, which led to a decrease in CCR7 [97]. Conversely, lncRNA CARLo-7, which was established as the only bladder-cancer-specific lncRNA in the CARLo cluster, is dramatically overexpressed in bladder cancer cells. Silencing of CARLo-7 repressed activation of JAK/STAT and Wnt/β-catenin pathways and, therefore, migration, invasiveness, and EMT were diminished [98].

### 4.5. Angiogenesis in Bladder Cancer

STAT3, besides being involved in EMT, participates in bladder cancer angiogenesis. Occludin is a protein that is a part of the tight-junction proteins family. This protein was overexpressed in bladder cancer tissue and was linked to the progression of cancer. Importantly, occludin facilitated angiogenesis by stimulating IL-8 secretion, mediated by STAT4. High levels of IL-8 triggered the phosphorylation of STAT3 and led to angiogenesis [99].

### 4.6. Impact of STAT3 on Bladder Cancer Cells’ Metabolism

STAT3 plays a significant role in bladder cancer metabolism. PLCε was found to be upregulated in bladder cancer and caused the stimulation of STAT3 phosphorylation, which regulates the transcription of LDHA and, as a result, affects glucose consumption and lactate production. Knockdown of PLCε in T24 cells attenuated STAT3 phosphorylation and resulted in LDHA expression, cell proliferation, glucose consumption, and lactate production downregulation [100]. The Gasdermin B protein, which is responsible, similarly to PLCε, for the regulation of pyroptosis in cells, was demonstrated to upregulate glucose metabolism via STAT3 activation. This interaction resulted in increased expression of LDHA, ENO2, HK2, and IGFBP3, which enhanced glycolysis [101]. It is worth mentioning that the deprivation of glutamine decreased the values of phosphorylated STAT3. What is more, the study implied that GLN is responsible for cell growth stimulation, mediated by STAT3. GLN not only regulates STAT3 by glutaminolysis and ATP supplementation but also through ROS level modulation in bladder cancer cell lines [102]. In addition, it was revealed that an elevated amount of RORC, with expression that is considered to be downregulated in bladder cancer, has probably led to the inhibition of cell proliferation and glucose metabolism, by suppressing the binding of STAT3 to the promoter of STAT3-mediated genes.

### 4.7. Drug Resistance in Bladder Cancer

Furthermore, RORC, via blocking STAT3, might have sensitized bladder cancer cells’ response to cisplatin [103]. Another study revealed that STAT3 can be a part of a process that facilitated doxorubicin resistance in bladder cancer. Phosphorylated protein was responsible for the recruitment of DNMTB3 to the promoter region of ESR1, which ended up with its hypermethylation and the downregulation of ESR1. Subsequently, this led to the downregulation of miR-4324. The microRNA, which decreased RACGAP1 protein levels, is supposed to be a tumor promoter and doxorubicin-resistance factor. Bladder cancer was proven to be resistant to radiotherapy [104]. Fractional irradiation enhanced the invasiveness, motility, and stem-like characteristics of bladder cancer cells. Responsibility for this effect was assigned to STAT3, which was previously reported to be significantly phosphorylated in irradiated cells [105]. Knockdown of STAT3 brought a loss of tumor formation in immunodeficient mice. It was suggested that excessive secretion of cytokines resulted in activation of the JAK/STAT pathway. Notably, it was demonstrated that the IL-6/STAT3 pathway is important to maintain stem-like properties [106]. Interesting findings were made during the concomitant utilization of Stattic and the chemotherapeutic agent approved for bladder cancer. The combined therapy of Stattic with one of the chemotherapeutics—gemcitabine, docetaxel, paclitaxel, or cisplatin—showed an additive effect in bladder cancer cell lines. This effect may find an application in the future, within a group of patients with chemoresistant tumors [90]. Besides, the combination of Stattic and Palbociclib (CDK4/6) inhibitor showed an additive effect in the T24 and UMUC-3 cell lines [90].

### 4.8. Role of STAT3 in Bladder Cancer Diagnostics

STAT3 may be considered a new potential biomarker. It was determined that its level correlates with a poor prognosis, but, due to conflicting reports, it is still disputed if the utilization of STAT3 as a new survival biomarker would be relevant [90,91]. However, it may prove to be useful in the differentiation of bladder cancer types, as the aforementioned urothelial type of bladder cancer was enriched in STAT3 [92]. Moreover, it Is suggested that the high expression of pSTAT3 is a strong predictor of the basal type of urothelial bladder cancer [91]. Interestingly, the value of pSTAT3 is elevated in high-grade NMBIC in comparison to low-grade, which indicates a possibility to detect more aggressive cells in NMBIC [91].

## 5. Role of STAT3 in Upper Tract Urothelial Carcinoma

UTUC is a rare type of neoplasm. It is defined as urothelial lining cells’ malignancy within the renal calyces, renal pelvis, ureter down, and ureter orifice. It comprises 5% of urothelial cancers and 10% of renal cancers. Two times as frequent, urothelial pelvicalyceal cancer is diagnosed as urothelial ureter cancer [107]. UTUC is associated with FGFR3 signaling, the papillary-luminal phenotype, and a T-cell depleted environment [108]. It is challenging to treat a multifocal disease, and relapses are often reported after initial therapy. Overall, 25% of patients present with metastasis when diagnosed [107]. The standard treatment in the case of high-risk UTUC remains radical nephroureterectomy. Surgery is combined with platinum-based chemotherapy, which improved the prognosis in comparison with radical treatment solely. Furthermore, platinum agents are also utilized in metastatic cancer at the first line [108].

STAT3 was proven to be a cancer-promoting factor and a valuable prognostic marker in UTUC. Higher values of STAT3 were reported in ureteral UTUC than in renal UTUC. Surprisingly, no differences in levels of protein were found in muscle-invasive and non-muscle-invasive cancer [109]. The expression of STAT3^Tyr705^ in the upper urinary tract was similar to that in the lower urinary tract, despite the diversity of these two cancer types in clinical features. Increased phosphorylation of STAT3 may be connected to the invasiveness and degree of histological differentiation [110].

Investigations demonstrated that elevated amounts of STAT3^Ser727^ were marked in 52% of patients. Higher values were identified with lower recurrence survival and cancer-specific survival. Patients with increased STAT3^Ser727^ were linked to a significantly poorer prognosis, higher cancer recurrence rate, and lower cancer-specific survival. Elevated expression of STAT3^Ser727^ in UTUC tissues, within the advanced cancer stage group of patients, was significantly associated with an advanced cancer stage and poor prognosis [111]. Another study demonstrated that STAT3 expression in the nucleus was connected to disease progression and lower cancer-specific survival. The results were similar to the aforementioned investigation, in that high STAT3 levels indicate UTUC progression, and the risk of exacerbation of the disease is considerably higher in the advanced stage group [109].

STAT3 is demonstrated to be a viable biomarker for invasiveness of cancer and metastasis, and it may provide a prediction of the need for more aggressive treatment. Interestingly, it was indicated that STAT3 could be a possible therapeutic target in UTUC [111].

## 6. Role of STAT3 in Renal Cell Carcinoma

### 6.1. Description and Epidemiology of Renal Cell Carcinoma

RCC is the deadliest urological cancer. Five-year survival rate prognosis is esteemed at 76%, although it decreases to 12% in the late stages [112]. In the year 2020, 432,288 new incidence cases and 179,368 death cases were reported. Hence, RCC was ranked as the sixteenth-most-common neoplasm in the world [1]. RCC can be distinguished according to the histological subtypes. In total, 90% of RCCs are of the clear cell carcinoma, papillary, and chromophobe histological subtypes. The ccRCC appears to be the most common and aggressive histological subtype, whereas a decreasing incidence of the remaining subtypes is reported, respectively [112].

### 6.2. Role of IL-6 in Renal Cell Carcinoma and STAT3 Levels

IL-6 was supposed to be an important autocrine growth factor of RCC. This cytokine proved to be detectable in renal carcinoma cell lines and genuinely stimulated the proliferation of the cancer cells [113]. However, it is still hard to say whether IL-6 is an autocrine growth factor, due to contradictory data [114]. Further investigations demonstrated that IL-6 stimulates proliferation via the activation of STAT3 [113]. Aberrantly phosphorylated STAT3 was observed in renal carcinoma, with notably elevated expression in the metastatic stage. Moreover, STAT3 was an indicator of poor prognosis and metastasis [115]. Interestingly, the number of tumors with activated STAT3 was similar in clear cell and papillary subtypes (57–59%), while a decreased number of cases were found within the chromophobe subtype (33%). Furthermore, STAT3 is reported to be a key player in clear cell carcinoma, as it upregulates 16 out of 32 genes, when expression was evaluated. A minor quantity of genes were upregulated in the papillary and chromophobe subtypes (10 and 7 genes, respectively). The MMP9, BIRC5, and BCL2 genes were notably upregulated, whereas the FOS gene was downregulated [114]. STAT3 signaling pathways, their regulators, their downstream targets, and the effects of their activation are shown in Figure 4.

### 6.3. Impact of STAT3 on Survival, Apoptosis, and Angiogenesis of Renal Cell Carcinoma

Inhibition of STAT3 leads to induction of apoptosis, reduction of cell viability, and proliferation in renal cancer cell lines. Suppression of the transcription factor also downregulated Bcl-2 levels, although levels of Bcl-xL and Mcl-1 remained unchanged. Additionally, angiogenesis was suppressed concomitantly with STAT3 inhibition [93]. Investigations demonstrated increased phosphorylation and nuclear localization of STAT3 during hypoxic conditions in the Caki I cell line. pSTAT3 was observed to increase the stability of HIF-1α by delaying protein degradation and accelerating its synthesis. The interaction of two proteins enhanced the expression of VEGF [116]. Notably, HIF-1α and VEGF are often upregulated in the ccRCC due to VHL gene mutation and are involved in the process of angiogenesis [117]. Inhibition of STAT3 suppressed VEGF expression, regardless of whether the VHL gene mutation eventually led to reduced tumor angiogenesis. This fact might find an application in the future [116].

A well-known marker of acute kidney injury HAVCR/KIM1 is also associated with elevated levels of pSTAT3. Overexpression of this marker was spotted in 60% of the ccRCC. HAVCR/KIM1 after cleavage by metalloproteinases triggers IL-6 secretion. This results in increased STAT3 phosphorylation and HIF-1α levels. An increased level of activated STAT3 also leads to enhanced expression of GLUT-1 and VEGF genes. Given that renal cell carcinoma is an angiogenic-rich neoplasm, it might be another potential pathway of cancer progression. RCC is considered a highly invasive cancer, as it is speculated that 33% of cases become metastatic [118], especially within the lungs, bones, and brain [119]. Investigation on G3BP1 revealed that IL-6 stimulates invasiveness and migration of RCC cells through STAT3 activation. G3BP1 was reported to be upregulated and mediate the process of STAT3 activation. Moreover, inhibition of G3BP1 decreased the STAT3 activation and, as a result, alleviated tumor growth and metastasis were observed not only in vitro but also in orthotopic xenografts [120].

### 6.4. EMT in Renal Cell Carcinoma

Additionally, the change in EMT markers was spotted, indicating that STAT3 promotes the metastatic process. EZH2 was discovered to be overexpressed in RCC cells and to regulate the proliferation and invasive potential of RCC cells. Subsequently, it was determined that this effect is exerted by STAT3 activation. In addition, EZH2 increased MMP2 expression, which is responsible for extracellular matrix degradation and related to tumor invasion and metastasis [119]. Low levels of vitamin D3 were associated with increased stimulation of the IL-6/STAT3 pathway. The treatment of cell lines with calcitriol almost completely decreased STAT3 phosphorylation. This data suggested that calcitriol can block the EMT process through STAT3 inhibition, at least partially [121]. Studies conducted on cancer-associated fibroblasts exposed the stimulatory effect on the migration of RCC cells. Increased amount of kynurenine due to TDO overexpression contributed to the activation of aromatic hydrocarbon receptors in renal cancer cells and of STAT3 [122]. Surprisingly, normal fibroblasts were also associated with increased cell migration in the ccRCC. Fibroblasts were reported to promote renal cancer cells to secrete IL-6 and phosphorylate STAT3. However, they did not demonstrate facilitation of the EMT. Nevertheless, the elevated expression of MMP2 was marked [123].

### 6.5. Drug Resistance in Renal Cell Carcinoma 

Cancer-associated fibroblasts were also revealed to have an impact on drug resistance to sunitinib and sorafenib in renal cancer cells [122]. IL-6 was linked with doxorubicin resistance in RCC cells, which was triggered by STAT3 phosphorylation. Attenuated STAT3 activity by si-IL6 sensitized renal cancer cells to doxorubicin [124]. IFN-α is utilized as an immunotherapy agent in the treatment of metastatic or recurrent RCC. Nevertheless, resistance to this drug is reported. It was observed that phosphorylation of STAT3 was increased by IL-6 but also by IFN-α. IL-6 is regarded as a negative regulator of the antiproliferative effect of IFN-α. Further studies have shown that IL-6 inhibition induced the efficacy of IFN-α; thus, the addition of tocilizumab may overcome the resistance of renal cancer to IFN-α [125].

### 6.6. Impact of STAT3 on Immune System

An interesting fact is that STAT3 is suggested to be involved in the immune escape of RCC cells. Overexpression of miR-129-3p, which is usually downregulated, reduced STAT3 and PD-L1 values and inhibited the proliferation, invasion, and immune escape of RCC cells. Additionally, overexpressed microRNA led to the enhancement of cytotoxicity, cytokine secretion, and proliferation of CD8+ T cells. This process was regulated by lncRNA SNHG1, which inhibited the activity of miR-129-3p. This makes SNHG1 a potential treatment target due to the ability of immune escape regulation [126].

### 6.7. Role of STAT3 in Renal Cell Carcinoma Diagnostics

STAT3 is suggested to present diagnostic value, as the phosphorylated protein in S727 residue correlated with prognosis and recurrence free survival. This investigation implies that STAT3^S727^ would be a valuable biomarker for the improved stratification and follow-up of patients during the same stage of cancer and clinical score [127]. High expression of STAT3 mRNA was also associated with shorter overall survival, contrary to low expression. However, due to the lack of statistical difference in the study, mRNA expression might not be useful as a biomarker [128]. Treatment with mTKIs is a standard method in metastasis, although this therapy often results in many adverse effects such as stomatitis or reactions in the hands and feet [129,130]. In addition, a differentiated response efficacy to mTKIs is observed. Adverse effects and treatment success were associated with the distribution of STAT3 polymorphisms. Therefore, assessment of STAT3 polymorphism might be a significant predictor of therapy efficacy and susceptibility to adverse effects [131].

## 7. Role of STAT3 in Penile Cancer

Penile cancer is one of the most uncommon cancers in the world. In 2020, there were 36,068 new cases and 13,211 deaths due to PeCa worldwide [1]. Although the incidence in Europe remains at a relatively low level, of 1/100,000, PeCa seems to be a more serious problem in central or southern parts of the Americas, such as Brazil, where the incidence is 6–8 times higher [6]. Risk factors of PeCa include phimosis, HPV infection, and smoking. What is more, scientists reported that socioeconomic factors such as educational level or marital status may help in predicting the occurrence of this disease [132].

Much attention has been recently paid to the role of specific chemokines, which are capable of regulating STAT3 expression in penile cancer. It was shown that CCL20, CXCL13, and CXCL5 are highly expressed in PeCa tissues. Elevated levels of those were also noticed in patients’ serum preoperatively. Knockdown of CCL20, CXCL13, or CXCL5 gave the same results shown by a significant suppression of pSTAT3 level and inhibition of MMP2 and MMP9 expression. Consequently, the diminished proliferation, migration, and invasion of PeCa cells were observed. High preoperative serum levels of the aforementioned chemokines were associated with tumor progression and poor outcomes. That is why scientists suggest using CCL20, CXCL13, and CXCL5 as diagnostic and prognostic biomarkers [133,134,135].

SHCBP1 is a gene that physiologically takes part in the T cell proliferation and signaling in neural progenitor cells. Scientists reported its association with development of some cancers, where it acts through the STAT3 pathway. Mo et al. explored its role in PeCa. SHCBP1 was significantly expressed in PeCa tissues, compared to the control group. The correlation between HPV infection and SHCBP1 expression was denied. On the other hand, it was highly correlated with the grading, staging, and status of the lymph nodes. Researchers conclude that SHCBP1 may be used as a prognostic biomarker. In vitro and in vivo experiments clearly showed that SHCBP1 knockdown results in decreased the proliferation, migration, and invasion of PeCa cells, while forced activation of STAT3 reverses this process. All in all, SHCBP1 has the potential to be used not only as a biomarker but also as a target for future treatment strategies [136].

## 8. Role of STAT3 in Testicular Cancer

Testicular cancer is one of the rarest cancers in the world, with 74,458 new cases and 9334 new deaths in 2020 [1]. Nevertheless, it is the most common cancer among young men between the ages of 15 and 35. Incidence varies geographically—the highest rates are in Europe (8.0–9.0/100,000), and the lowest are in Asia and Africa (<1.0/100,000). The greatest risk factor is a history of testicular cancer on the opposite side [2].

HOXA10 is a transcription factor that regulates testicular development. Scientists investigated its role and mechanism of action in testicular cancer. HOXA10 has been shown to be expressed and localized in the nuclei of spermatocytes in normal tissues, while it is often dislocated in TGCT cells, both seminomas and non-seminomas. Although its antiproliferative properties have been proven by inhibiting the STAT3 pathway, the exact mechanistic explanation remains unknown. Researchers acknowledged that detecting HOXA10 levels may not be useful in diagnosis due to unchanged expression in TGCT. Nevertheless, this study may be the basis for further experiments [137].

## 9. STAT3 Inhibitors as Viable Targets

There seems to be many ways of disrupting STAT3 pathway activation. On the one hand, some of the aforementioned molecules may be described as direct STAT3 inhibitors. On the other hand, many of them seems to interfere STAT3 activation pathway indirectly, such as Phoenix Rising [47]. Here, we would like to summarize the compounds capable of inhibiting STAT3 directly, which have the potential to become more interesting for clinicians in the nearest future.

There are a few molecules that have an ability to modify TME, which seems to be crucial in PCa development. Highly concentrated LTF is capable of changing TME immunologically [41]. S32-201 inhibits MSC-selected PCa cells [13]. Galiellalactone not only downregulates MDSCs’ frequency but also decreases PCa cells’ resistance to docetaxel and enzalutamide [44,52,53].

Managing CRPC and chemoresistance are really problematic for clinicians. Galiellalactone, mentioned above, seems to be effective at breaking the chemoresistance. Similarly, FBP1 has the potential to intensify anti-PD-L1 treatment [48]. HepaCAM effectively decreases the proliferation, migration, and invasion of CRPC cells [30]. ALT sensitizes CSCs to cisplatin [50]. Metformin reverses and inhibits EMT, including EMT promoted by ENZ [25,34]. NDRG1 affects crucial steps in the STAT3 activation pathway, disrupting androgen-independent AR activation pathways, which may benefit CRPC patients [57].

Iridium(III)-benzofuran complex was found to be more toxic against DU145 cells than cisplatin and doxorubicin, with simultaneous lower toxicity to normal human cell lines [28]. ASC-J9 decreases STAT3 phosphorylation by sumoylation in PCa [69]. Calcitriol was described as an effective STAT3 inhibitor in both PCa and RCC [63,121]. SAM significantly reduces the STAT3 and pSTAT3 levels in PCa [39].

Fucoidan, atractylenolide, proscillaridin A, methyllucidone, polymethoxyflavone, astaxanthin, and compound 154 are natural molecules with an ability to disrupt the JAK/STAT3 pathway [32,35,72,73,81,82]. Other organic molecules or their derivatives have an ability to bind to STAT3 directly [76,77,78].

In the field of bladder cancer, Stattic significantly reduces the weight of tumor xenografts [90]. Tanshinone IIa effectively suppresses EMT by inhibiting STAT3 phosphorylation, whereas miR-4500 presents similar effects by base pairing [94,97]. Finally, RORC impairs STAT3 binding to STAT-3 mediated genes, leading to bladder cancer cells being sensitized to cisplatin [103].

On the other hand, HOXA10 in testicular cancer seems to be a promising target. Although its exact mechanism of action remains unknown, its STAT3 pathway inhibition was proven [137].

## 10. Discussion 

STAT3 appears to be a promising molecular target for the diagnosis and treatment of urological neoplasms. It is a point of convergence for many signaling pathways induced by cytokines, growth factors, and oncoproteins [11].

TME is important for disease progression. Studies have revealed that cancer-related fibroblasts are involved in enhancing the migratory properties of the prostate, kidney, and kidney cancer, by secreting several stimulating factors such as IL-6 or CCL3 that activate STAT3. TAMs were also involved in the migration, invasion, and EMT in PCa by secreting CCL5 [16,23,24,25,26].

Interestingly, STAT3 has been found to bind to the PD-L1 promoter in cancer cells. It is believed to be of key importance in regulating the immune response in TME. IL-6-rich tumors were more resistant to cytotoxic NK cells. JAK1 or STAT3 inhibition reduced the PD-L1 levels in PCa cells [27]. Moreover, the IL-6/STAT3/PD-L1 pathway has been shown to be involved in the promotion of EMT in bladder cancer [28]. Moreover, decreased levels of STAT3 and PD-L1 were associated with the inhibition of the proliferation, invasion, and immune escape of RCC cells [30].

STAT3 is able to induce chemoresistance in neoplastic cells, which can be a significant problem during treatment. STAT3-induced resistance to doxorubicin has been observed in RCC and bladder cancer [29,31]. In turn, it has been reported that the phenomenon of autophagy promotes docetaxel resistance in CRPC cells [51]. Moreover, CYP1B1 increases the resistance of CRPC cells to bicalutamide [49]. However, galiellalactone appears to be effective in breaking resistance not only to docetaxel but also to ENZ [52,53]. STAT3 inhibition led to PCSC sensitization to cisplatin in PCa and bladder cancer and docetaxel in PCa [32,103]. On the other hand, ALT has the ability to sensitize neoplastic stem cells to cisplatin [50], while the inhibition or knockdown of STAT3 led to an increase in the sensitivity of PCa cells to radiation [55].

There are also promising combination therapies. STAT3 inhibition and TLR9 stimulation showed significant changes in TME in PCa [45]. A JAK or STAT3 inhibitor coupled with the PD-L1 antibody increases the susceptibility of CRPC cells to cytotoxic NK cells [36]. Comprehensive treatment of radio- and chemotherapy increases apoptosis of PCa cells [55]. On the other hand, the combination of benzofuran-iridium(III) turned out to be more toxic to DU145 cells than cisplatin or doxorubicin [28]. Bladder cancer turned out to be sensitive to combinations of the well-known Stattic preparation and complexes—gemcitabine, docetaxel, paclitaxel, or cisplatin [90].

Several attempts have been made to use drugs unknown to PCa chemotherapy but widely used in other branches of medicine. EA has been shown to have anti-proliferative properties [66]. Capz can inhibit STAT3 phosphorylation and nuclear translocation [67]. Metformin inhibits the COX-2/PGE-2/STAT3 pathway, directly inhibits STAT3 when it is highly concentrated [34], and has the ability to reverse ENZ-promoted EMT [53].

The features of RNA molecules appear to be useful in targeting STAT3. MAGI2-AS3 acts as a sponge for another lncRNA that activates STAT3 in PCa [58]. Transfection of miRNA-583 into PCa cells inhibits the JAK/STAT3 pathway [59]. LINC00467 induces polarization from prostate M1 macrophages to M2 macrophages, thus activating the STAT3 pathway [60]. miR-129-3p stimulates the immune system in RCC, resulting in the inhibition of proliferation and invasion [126].

Risk factors such as a high-fat diet and obesity have been shown to have a significant impact on STAT3 activation. PA increases the levels of mRNA and STAT3 protein. Similarly, LDL cholesterol increases pSTAT3 levels through increased JAK1 and JAK2 phosphorylation [4,26]. Leptin has been shown to promote EMT by activating STAT3 [15]. Moreover, low vitamin D3 levels were associated with stimulation of the IL-6/STAT3 pathway in PCa and RCC. Calcitriol used in therapy reduced STAT3 phosphorylation [35,36]. STAT3 has been shown to affect cellular metabolism, through increased glucose consumption, and lactate production in bladder cancer, through increased transcription of LDHA, ENO2, HK2, and IGFBP3 [37,39].

Many researchers have recently focused on natural compounds that can benefit cancer patients. Many of the described complexes have proven to be effective in PCa. Some of them inhibit the JAK2/STAT3 pathway [72,73], while others have the potential to directly bind to STAT3, disrupting its dimerization and nuclear translocation [76,77,78]. There are also compounds capable of modifying TME, inhibiting PD-L1 expression [79], and forcing the TAM phenotype to change from M2-like to M1-like [12]. In the field of bladder cancer, Tanshinon IIa has been reported to inhibit EMT [94].

STAT3 can also be used as a valuable biomarker of tumorigenesis. STAT3 levels were elevated in high-grade NMIBC tumors. The usefulness for distinguishing the type of neoplasm has also been found [38,40]. Increased levels of STAT3^Ser727^ were associated with a significantly poorer prognosis, a higher cancer recurrence rate, and a lower cancer-specific survival in the UTUC. This study suggests that STAT3^Ser727^ may also be a valuable biomarker for better stratification and follow-up of patients with the same cancer stage and the same clinical scale in RCC [33,41,42]. In the field of PCa, lower levels of STAT3^Ser727^ and STAT3^Tyr705^ in the stromal and nuclear compartments have been observed in neoplastic tissues rather than in non-neoplastic tissues [83]. Another team reported that STAT3^Ser727^ and STAT3^Tyr705^ levels in prostate epithelial cells negatively correlate with poor prognosis [84]. Moreover, detection of specific post-translational STAT3 modifications such as phosphorylation, acetylation, and glutathionylation can be a biomarker in the prevention or differentiation of PCa [68].

## 11. Conclusions

The main aim of this review was to collect and discuss the most important recently published papers focusing on the role of STAT3 in oncological urology. Considering the field of urological cancer as a significant issue, we hope that the above information will encourage readers to explore it.

STAT3 acts multidimensionally by shaping TME, EMT, resistance, etc., and its precise role in inducing (and suppressing) cancerogenesis has not yet been studied. Although it seems difficult in clinical practice to consider all these processes together, they should be taken into account in laboratory studies.

Recently published papers seem to focus heavily on immunology. TME PCa was thoroughly analyzed. The relationship between M1-like and M2-like phenotypes macrophages, MDSCs, and other components of the tumor microenvironment can be extrapolated to other urological neoplasms. An interesting approach may be to study these correlations in the context of bladder cancer or RCC.

The article discusses some of the TME regulating factors. Galiellalactone appears to act broadly, so it is worth exploring its properties in more depth. FBP1, the prominent glycolysis enzyme, has the potential to contribute to combination therapies by overcoming resistance to ani-PD-L1 treatment. The attraction of immune cells by stimulation of TLR9 also looks like an innovative but still not fully described way of weakening emerging anti-cancer immunity.

## Figures and Tables

**Figure 1 cells-11-03024-f001:**
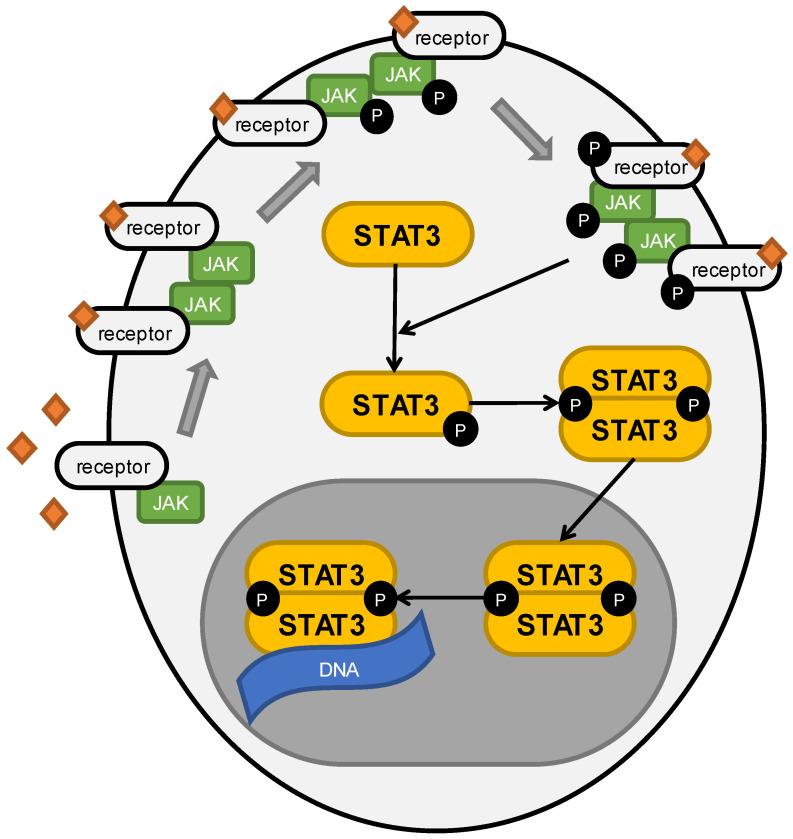
Schematic and simplified depiction of STAT3 pathway activation. Orange diamonds present ligands’ molecules, and black circles marked with “P” present phosphate group. Figure presents the process of receptor dimerization due to their activation by ligands, JAK transphosphorylation, receptors’ phosphorylation, STAT3 phosphorylation, STAT3 dimerization, translocation to the nucleus, and, finally, its binding to DNA.

**Figure 2 cells-11-03024-f002:**
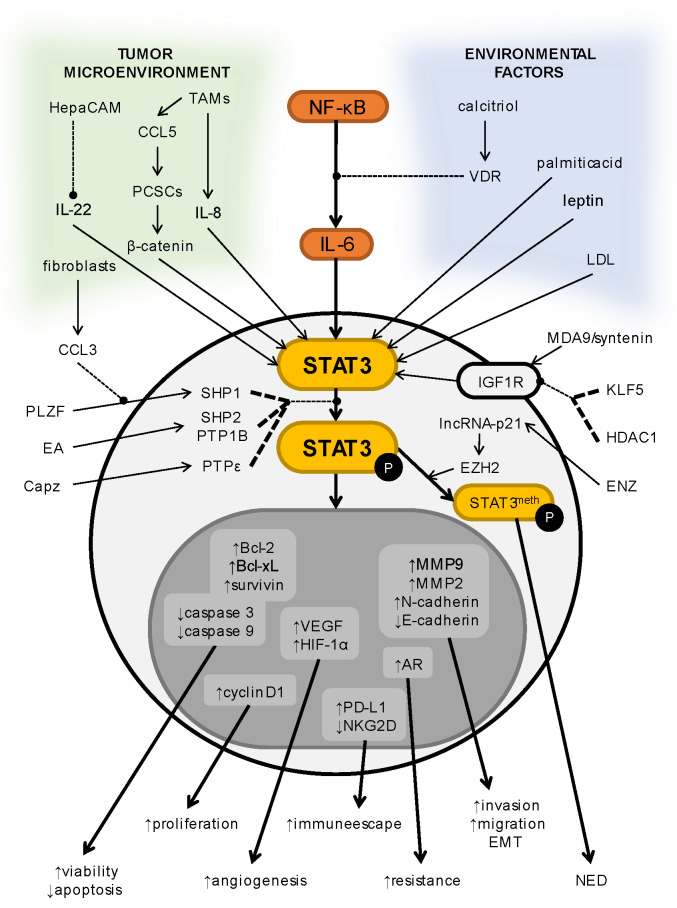
Schematic and simplified depiction of selected pathways and factors regulating STAT3 expression, counting its downstream target proteins, with the consequences of their overexpression in prostate cancer. Dashed lines ending with a dot show inhibition; arrows show stimulation.

**Figure 3 cells-11-03024-f003:**
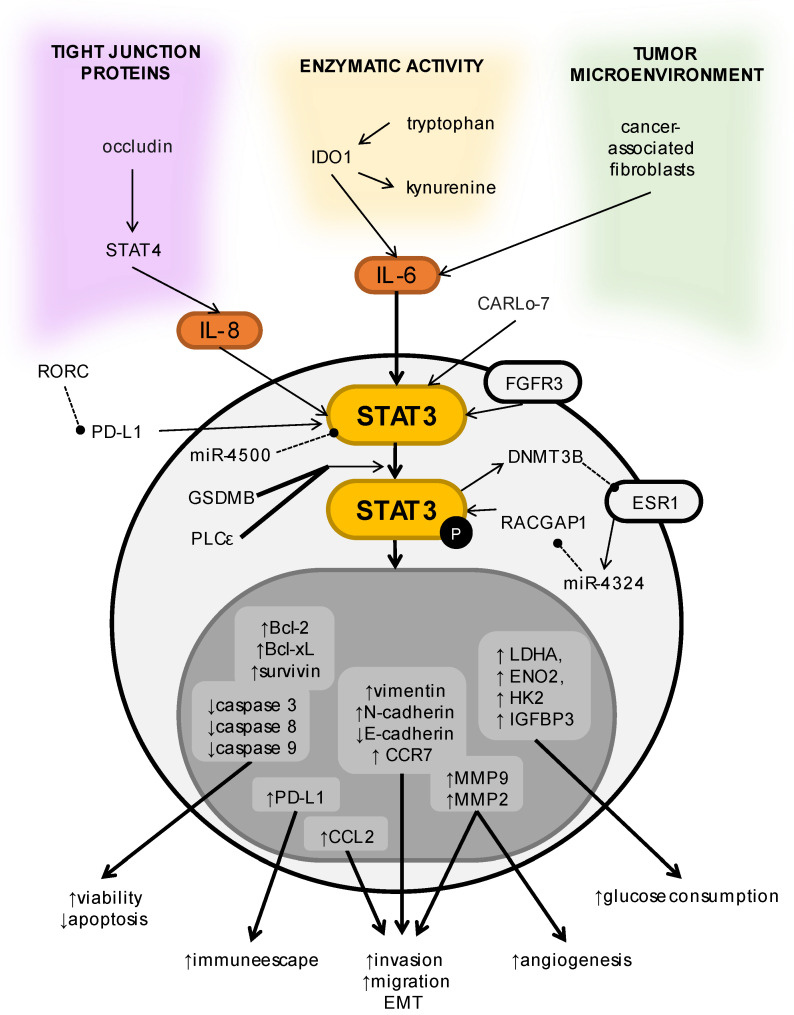
Schematic and simplified depiction of selected pathways and factors regulating STAT3 expression, counting its downstream target proteins, with the consequences of their overexpression in bladder cancer. Dashed lines ending with a dot show inhibition; arrows show stimulation.

**Figure 4 cells-11-03024-f004:**
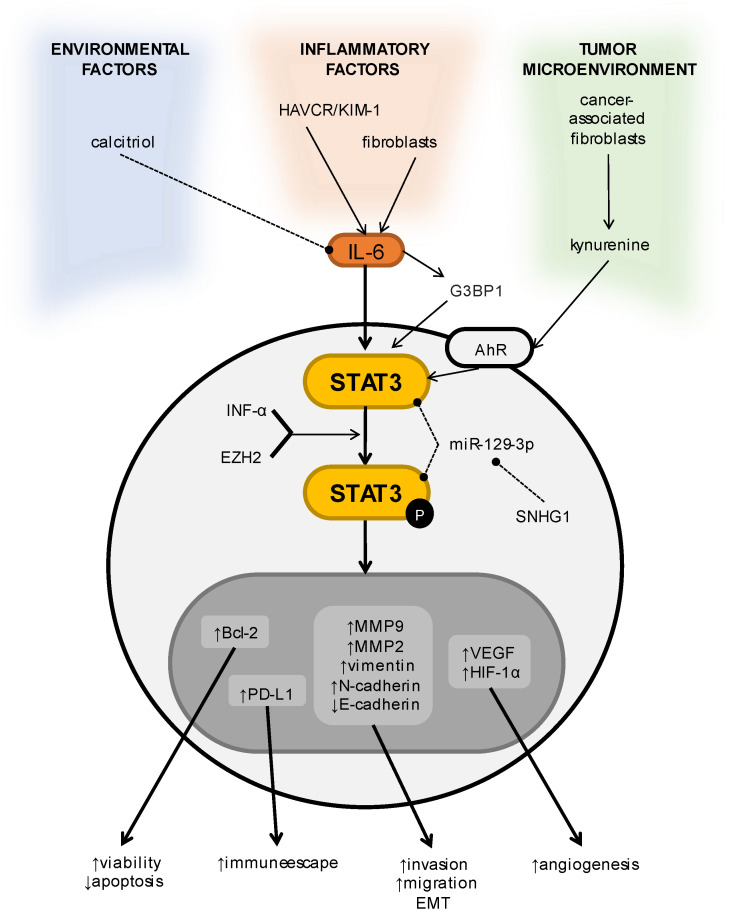
Schematic and simplified depiction of selected pathways and factors regulating STAT3 expression, counting its downstream target proteins, with the consequences of their overexpression in renal cell carcinoma. Dashed lines ending with a dot show inhibition; arrows show stimulation.

## Data Availability

Not applicable.

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
