# Peer review of "STAT3 and Its Pathways’ Dysregulation—Underestimated Role in Urological Tumors"

_cells, 2022, doi:10.3390/cells11193024_

Round 1

Reviewer 1 Report

The authors present a broad overview on the role of STAT3 in different urological tumor types. Although the review is potentially interesting to a broader readership several concerns were raised:

Major points:

1.      In the part on the specific role of STAT3 in the different urogenital cancer types more structure, such as subheadings for distinct paragraphs would be recommended (eg for part on fibroblasts, TAMs, immunological checkpoints etc.). At the moment especially chapter 4 is difficult to read. In addition, it is not always clear whether it is referred to STAT3 in tumor cells or immune cells.

2.      In the introduction it was mentioned that STAT3 has a prooncogenic role in many different cancers. However, in some instances STAT3 can also act as a tumor suppressor, also in prostate cancer (eg Pencik J et al, Nature Comm. 2015). 

3.      STAT3 inhibitors as “viable targets” are suggested several times in the review. A specific section on STAT3 inhibitors would be useful information for the readers.

4.      PC3 cells do not express STAT3 (eg Poria D et al, JBC 2021), this information might be of interest for the reader for the parts where this model system was mentioned. Eg Line 391: “…xxx is also cytotoxic to PC3 cell line through JAK1/STAT3 pathway deactivation” ?

5.      Chapter 9, lines 741-744: Please be more specific on STAT3 here.

6.      Discussion: First paragraph is a bit out of focus of this review. Second and part of third paragraph are highly redundant with introduction and should be omitted or shortened. In the discussion the similarities and differences of the role of STAT3 in the mentioned urological cancer entities should be stressed more.

7.      A conclusion (with broader implications) is missing.

Minor points:

1- Abstract: in the list of the seven STAT members STAT4 is missing

2- Line 53: transcriptional factor

3- Numbering of chapters: chapter 2 is missing

4- Line 72: STAT4 is missing among the seven STAT members

5- Chapter 3: when explaining the pathway activating STAT3 please name the kinases

6- Line 87: the JAK/STAT3 pathway

7- Lines 92-96: quite redundant with the introduction part above

8- Please refer to the figures within the text

9- Line 176: immunological immunity?

 - Line 183: Sudden change from MDSC to MSC – new paragraph or explain connection

 - -  Line 219: please correct “carried experiments”

 -  The authors state that there is a role for pSTAT3Ser727 in UTUC; was pSTAT3Tyr701 also addressed in this work?

 -  Line 553: excessive secretory of cytokines

 -  Lines 557-558: does this refer to in vivo models or patients? Please clarify

 -  Chapter 7: abbreviation of interferon (IFN)

 -  Line 763: STAT3 acts as…

 -  Line 814: which post-translational modifications?

Reviewer 2 Report

The review by M Golus et al gives a very comprehensive (and dense) overview of the role of the transcription factor STAT3 in urological tumors. The review contains a very large amount of references, which meets the ambition of this review as announced by the authors.

I congratulate the authors for the enormous amount of work in gathering information and writing this review, which appears very complete.

However, the density of information makes the reading very painful, with a succession of pages without chapters and an impression of a catalog. I therefore invite the authors to structure the chapters into sub-chapters / paragraphs to lighten the text and allow the reader to find specific information more easily.

The text has many abbreviations that should be listed in a specific section.

The figures are also very dense - and are not referenced in the text. I suggest that the authors use color codes to specify the nature of the molecules presented. A figure presenting STAT3 signalling pathways (phosphorylation and dimerization) and regulatory pathways (phosphatases, lcnRNA, miRNA, ...) should be added to the manuscript.

The authors mention several times in their text the M1 and M2 macrophage subtypes. This classification is only valid for the mouse model. The authors should therefore modify the text accordingly.

The text should also be proofread conscientiously to avoid phrases that are quite surprising, such as "immunological immunity" lane 176.

A number of inaccuracies are found throughout the text; as examples, I have noted

Lane 175: “MDSC levels”, perhaps instead of frequency?

Lane 181: "bifunctional and immunostimulatory combination" bifunctional refers to what?

L205: “NK cells were tried to manage CRPC” what does it mean?

Similarly, authors should avoid successive and redundant sentences, such as lines 212 and 214 and 246 and 247.

Round 2

Reviewer 1 Report

The authors have sufficiently addressed many of the comments. However, some issues still need to be improved.

-) Please correct the typos and improve the English of the new paragraphs

-) Chapter 2, line 141: to mention JAK kinases in general is sufficient (as one ligand does not activate all 4 JAK family members at the same time)  

-) Line 143: rather "or" instead of "and"

-) Figure 1, legend: why is referred to chemokines here (lines 174, 175)? As it is a very general scheme "ligand" would be better

-) The last paragraph of the conclusion is dispensable

Reviewer 2 Report

I thank the authors for thoroughly editing the manuscript and for following my recommendations and suggestions.
The review has been greatly improved and is now easy to read.

Author Response

Letter with a reply to the Reviewer No. 2 Report

We thank the reviewer for encouraging comments, and appreciate the insightful comments and suggestions that have helped us significantly improve the quality of this review.

Kind regards,

Author's